# Helmet Wearing State Detection Based on Improved Yolov5s

**DOI:** 10.3390/s22249843

**Published:** 2022-12-14

**Authors:** Yi-Jia Zhang, Fu-Su Xiao, Zhe-Ming Lu

**Affiliations:** 1School of Information Science and Engineering, Zhejiang Sci-Tech University, Hangzhou 310018, China; 2School of Aeronautics and Astronautics, Zhejiang University, Hangzhou 310027, China

**Keywords:** helmet wearing states, small target detection, YOLOv5s

## Abstract

At many construction sites, whether to wear a helmet is directly related to the safety of the workers. Therefore, the detection of helmet use has become a crucial monitoring tool for construction safety. However, most of the current helmet wearing detection algorithms are only dedicated to distinguishing pedestrians who wear helmets from those who do not. In order to further enrich the detection in construction scenes, this paper builds a dataset with six cases: not wearing a helmet, wearing a helmet, just wearing a hat, having a helmet, but not wearing it, wearing a helmet correctly, and wearing a helmet without wearing the chin strap. On this basis, this paper proposes a practical algorithm for detecting helmet wearing states based on the improved YOLOv5s algorithm. Firstly, according to the characteristics of the label of the dataset constructed by us, the K-means method is used to redesign the size of the prior box and match it to the corresponding feature layer to increase the accuracy of the feature extraction of the model; secondly, an additional layer is added to the algorithm to improve the ability of the model to recognize small targets; finally, the attention mechanism is introduced in the algorithm, and the CIOU_Loss function in the YOLOv5 method is replaced by the EIOU_Loss function. The experimental results indicate that the improved algorithm is more accurate than the original YOLOv5s algorithm. In addition, the finer classification also significantly enhances the detection performance of the model.

## 1. Introduction

The construction site environment is complex; objects, as well as operators may fall from a height at any time. Injuries due to accidents can be effectively decreased by wearing safety helmets. However, tragedies resulting from inadequate supervision of the construction system and insufficient safety awareness of the workers occasionally occur. Therefore, supervising the wearing of safety helmets through a helmet wearing detection algorithm has high practical value.

The early studies primarily used manual feature extraction to detect the wearing of helmets. The mainstream research idea is to locate the position of the pedestrian using the HOG feature, C4 algorithm, and other methods [1,2,3,4] and then identify the characteristics of the helmet in the head area, such as the color, contour, and texture [5,6,7]. Finally, SVM and other classifiers were used to complete helmet detection [8,9]. Figure 1 shows the four main implementation steps of this kind of algorithm: pre-processing, Region Of Interest (ROI) selection, feature extraction and detection or classification. Because of the simple structure of the algorithm, the traditional algorithm has less computational requirements and a faster detection speed.

However, there is still a gap between the detection effect of the traditional algorithm and the practical application requirement of high precision. The effectiveness of helmet wearing detection is poor under the traditional algorithm, especially when the frames are under the influence of illumination and angle variability. Convolutional neural networks are frequently employed in target detection in various disciplines [10] due to their great feature extraction capabilities with the emergence of deep learning methods [11,12,13,14]. Scholars have successively applied RCNN, fast RCNN, SSD, YOLO, and other algorithms to the research of helmet wearing detection [15,16,17,18,19,20]. Among them, the SSD and YOLO algorithms have higher accuracy as one-stage algorithms, while YOLO has a higher detection rate on this basis, which makes the YOLO algorithm stand out in the research and application of helmet wearing detection [21,22,23]. In order to better explain the optimization process of the helmet wearing detection algorithm, we provide Figure 2.

However, the majority of the YOLO series of algorithms for detecting helmet wearing perform two-class tests on the dataset SHWD, which merely determines whether a helmet is being worn or not. It is difficult for those algorithms to meet the comprehensive helmet wearing state detection under complex conditions, and there is little space for further research.

In this paper, an algorithm based on the improved YOLOv5s algorithm is proposed for detecting the wearing states of safety helmets. The specific contributions are as follows: (1) Different from the existing datasets, we construct a six-class helmet wearing dataset, which aims to distinguish the different states of helmets in the construction scene and improve the feature extraction accuracy and detection performance of the whole model. (2) A small target detection layer is added to the YOLOv5 network, and the anchor size is revised in accordance with the new detection layer and the dataset we constructed. In addition, the attention mechanism is introduced to the backbone network of YOLOv5s, and its initial CIOU_Loss is replaced with the EIOU_Loss function. (3) Using our dataset to train the improved YOLOv5 algorithm, we can obtain a model that can accurately identify the wearing of safety helmets.

This paper is organized as follows: Section 2 presents the YOLOv5s algorithm and some improved techniques of this paper. The experimental process and analysis of the improved YOLOv5s algorithm are elaborated in Section 3, such as the experimental setup, dataset acquisition, training and test results, and ablation experiment. Section 3 also compares the improved algorithm with some current helmet wearing detection algorithms to further show the experimental effect. The research of this paper is finally concluded in Section 4, which also suggests the future work.

## 2. Methodology and Improvement

### 2.1. YOLOv5s Algorithm

The input, backbone, neck, and prediction output make up the YOLOv5s algorithm [24], and its framework is shown in Figure 3. The backbone network is the feature extraction network, which mainly includes the CBS module, CSP module, and fast spatial pyramid pooling (SPPF) module. The CBS module is the combination of the convolutional module, batch normalization module, and activation function, named SiLu. The CSP structure divides the original input into two branches for the convolution operations, so that the number of channels is halved, and concats two branches, so that the input and output of the CSP are the same size. In other words, the CSP allows the model to learn more features. Moreover, the CSP includes CSP1_X and CSP2_X, the main difference between them being that there is a residual module in CSP1_X, and CSP2_X corresponds to the CBS module. The residual structure can increase the gradient value of the back-propagation between layers, avoiding the gradient loss caused by network deepening, so that features with finer granularity can be extracted without worrying about network degradation; the SPP structure can convert any size of feature map into a fixed-size feature vector. The SPPF structure used in YOLOv5 replaces the original parallel MaxPool of the SPP structure with a serial MaxPool, making the SPP structure more efficient.

The neck is the feature fusion network that combines the top-down and bottom-up feature fusion techniques in order to more effectively incorporate multi-scale features extracted from the backbone network before transferring them to the detection layer. After the non-maximum suppression and other post-processing operations, a large number of redundant prediction frames are eliminated. Finally, the prediction category with the highest confidence score is output, and the frame coordinates of the target position are returned.

Based on the strong detection and discrimination ability of the YOLOv5s algorithm, in this paper, to detect the wearer of a helmet in a variety of situations, we improved and adjusted the prior frame and loss function of the algorithm to detect the wearing states of safety helmets in various scenarios. In order to more effectively detect distant and dense targets, we also added a small target detection layer to the framework. In the selection of the loss function, although CIOU_Loss adopted by the YOLOv5s algorithm fully considers the overlapping area, center point distance, and aspect ratio of the boundingbox regression through previous improvements, we adopted EIOU_Loss with a better aspect ratio and stronger robustness as the loss function of the algorithm. Finally, we added an attention mechanism to the network to improve its detection capabilities as a whole.

### 2.2. Redesign the Prior Anchor Frame

The prior anchor frame data of the original YOLOv5s algorithm is calculated according to the characteristics of the eighty-class dataset of COCO. In order to make the YOLOv5s algorithm work better in our helmet wearing state detection research, we rebuilt the prior box size in YOLOv5s’s algorithm using the K-means approach in accordance with the length-width ratio and other elements of our helmet dataset, so that the prior box size is more consistent with our dataset. Specifically, we first selected the number of anchors *k* (9 or 12 in this paper) and initialized *k* anchor boxes. For the bounding box of each sample in the dataset, we calculated its Intersection Over Union (IOU) with each anchor box, classified the sample into the anchor box with the largest IOU, and recalculated and updated each anchor box. We repeated this until no anchor boxes changed, and finally we realized the clustering of the anchors. Table 1 lists the prior anchor frame sizes before and after modification. It should be noted that we added a small object detection layer to the original YOLOv5s algorithm framework, so we correspondingly added a prior anchor frame under the small target detection scale.

Table 2 illustrates the comparison of the algorithm’s convergence speed before and after anchor modification. It can be seen from the table that the YOLOv5s algorithm with the redesigned anchor has a faster convergence speed, which significantly increases the training efficiency of the model. In addition, the mean Average Precision (mAP) of the modified anchor algorithm also increased by roughly 1%.

### 2.3. Add a Small Target Detection Layer

In order to better apply this algorithm to the actual scene, the initial network structure of YOLOv5s was modified in this paper to solve the problem of the YOLOv5s algorithm having an insufficient effect in detecting long-distance and small targets in the study of helmet wearing status. Specifically, on top of the initial three detection layers, we added a small target detection layer to allow the model to pull feature information from deeper networks and enhance its capacity to recognize small objects. Figure 4 displays the network structure of the enhanced model.

We selected the mean Average Precision when the IOU is 0.5 (mAP@0.5) and the mean Average Precision when the small target area is less than $32^{2}$ (mAP@small) in the COCO evaluation index system to evaluate the improvement result. Table 3 shows that the accuracy was improved when the small target detection layer was added, and the model’s capability to recognize small objects was significantly improved.

### 2.4. Adopt EIOU_Loss

After the improvement of IOU_Loss to CIOU_Loss, the loss function of the YOLO algorithm was able to comprehensively consider the overlapping area, center point distance, and aspect ratio of bounding box regression. Furthermore, some scholars split the loss term of the aspect ratio into the difference between the predicted width and height and the minimum external frame width and height based on CIOU_Loss [25,26], accelerating the convergence and improving the regression accuracy. The more efficient loss function is EIOU_Loss. Next, we will introduce and compare the two loss functions.

First, the penalty term of CIOU_Loss is shown in Equation (Equation 1).
(1)$$R_{CIOU}=\frac{\rho^{2}\left(b,b^{gt}\right)}{c^{2}}+\alpha v$$
where *b* and $b^{gt}$ represent the center points of the prediction box and the ground-truth box, respectively, and ρ represents the Euclideandistance between two center points. *c* indicates the diagonal distance of the minimum closure area that can contain both the prediction box and the ground-truth box. α is a weight function, and *v* is used to measure the similarity of the aspect ratio. α and *v* are defined by Equations (2) and (3), respectively.
(2)$$\alpha =\frac{v}{(1-IOU)+v}$$
(3)$$v=\frac{4}{\pi^{2}}{\left(arctan\frac{w^{gt}}{h^{gt}}-arctan\frac{w}{h}\right)}^{2}+\alpha v$$

The complete CIoU_Loss function is defined in Equation (Equation 4).
(4)$$L_{CIOU}=1-IOU+\frac{\rho^{2}\left(b,b^{gt}\right)}{c^{2}}$$
where $C_{w}$ and $C_{h}$ are the width and height of the minimum bounding box covering the two boxes.

CIOU_Loss considers the overlapping area, center point distance, and aspect ratio of bounding box regression. However, it reflects the difference of the aspect ratio through *v*, rather than the real difference between the width and height and their confidence, so it sometimes hinders the effective optimization similarity of the model. The penalty term of EIOU_Loss is used to calculate the length and width of the target frame and anchor frame, respectively, by separating the influence factor of the aspect ratio on the basis of the penalty term of CIOU_Loss. The loss function includes three parts: overlapping loss ($L_{IOU}$), center distance loss ($L_{dis}$), width and height loss ($L_{asp}$). The first two parts continue the method in CIOU_Loss, but the width and height loss directly minimizes the difference between the width and height of the target frame and anchor frame, making the convergence speed faster. The EIOU_Loss function is defined by Equation (Equation 5).
(5)$$\begin{array}{cc} L_{EIOU} & =L_{IOU}+L_{dis}+L_{asp}\\ \; & =1-IOU+\frac{\rho^{2}\left(b,b^{gt}\right)}{c^{2}}+\frac{\rho^{2}\left(w,w^{gt}\right)}{C_{w}^{2}}+\frac{\rho^{2}\left(h,h^{gt}\right)}{C_{h}^{2}} \end{array}$$

As shown in Table 4, by replacing CIOU_Loss with EIOU_Loss in the YOLOv5s algorithm, the mAP of the model is increased by 1.2%. Moreover, the model’s convergence speed is also accelerated by the use of EIOU_Loss.

### 2.5. Increase Attention Mechanism

Based on the series of improvements mentioned above, we added an attention module to the network in order to further enhance the model’s capacity for detection and force the network to focus more on the target to be detected. Our specific actions mainly involved two methods: one is to insert the attention module into the tenth layer of the backbone of the YOLOv5 model (such as SE, CBAM, ECA, and CoordAtt), and the other is to replace all CSP modules (Layers 3, 5, 7, and 9) in the backbone of the YOLOv5 model with our attention modules (such as C3SE). In order to select the attention mechanism that is most suitable for the helmet wearing states detection network in this research, we trained SE, C3SE, CBAM, ECA, CoordAtt, and Transformer, respectively [27,28,29,30,31,32,33], and the findings are displayed in Table 5. The detection performance of the YOLOv5s algorithm was improved after introducing the attention module. Moreover, CoordAtt performed best with this algorithm.

Moreover, we also tried to introduce the lightweight module Ghost in the experiment. Table 5 indicates that, while Ghost reduces the model’s weight, some accuracy is sacrificed in the process.

## 3. Experiment and Analysis

### 3.1. Experimental Setup

In our experiments, the operating system was Linux, the CPU was a AMD Ryzen 9 5950X 16-Core Processor 3.40 GHz, the GPU was a Tesla v100-sxm2-16GB, the framework was Pytorch, the batch size was set to 16, the epoch was set to 300 (the early stopping mechanism was also enabled), and the image size was 640 × 640.

### 3.2. Dataset

At present, there are few datasets on helmet wearing. The public dataset named SHWD only includes two cases of helmet wearing and pedestrians, which cannot reflect the various states of the helmet in the real construction scene completely. Therefore, this paper collected 8476 images using dataset selection, web crawling, and self-shooting, and then we annotated them by labelImg to build a dataset of six categories which includes not wearing a helmet (person), only wearing a helmet (helmet), just wearing a hat (hat_only), having a helmet, but not wearing it (helmet_nowear), wearing a helmet correctly (helmet_good), and wearing a helmet without the chin strap (not_fastened). A wide range of construction scenarios were included in the dataset created for this study, which can accurately reflect real construction scenarios. However, in the early images of helmet wearing, most helmets were only attached to the head, and there was no design for the chin strap. In addition, it is difficult to judge whether a person is wearing a helmet correctly when he/she has a head covering or we have a remote view of his/her back. Therefore, the classification of “wearing a helmet (helmet)” in our dataset is more like a “suspicious” classification.

The dataset was split into a training set and a validation set at a 7:3 ratio. Table 6 lists the total number of target box annotations of each category in the dataset.

The sample of the six categories is shown in Figure 5. It is worth noting that a six-class dataset was constructed to better distinguish and recognize the use of helmets in the construction scene, and finer classification can also better improve the detection performance of the model. For example, the class “hat_only” can distinguish some situations better that interfere with the wearing of safety helmets (such as a worker wearing a baseball cap that is very similar to a safety helmet, as well as police and nurses at the construction site); the class “helmet_nowear” is intended to detect the situation where the helmet is held in the hand or there is a helmet in the environment, but it is not being worn. The above research can also pave the way for further image description research on this subject.

It is worth mentioning that the images in our test set were collected from a recent construction site, completely independent of the training set and validation set, which makes the test results more convincing. Figure 6 represents some samples of the training set and test set.

### 3.3. Training Results

The improved YOLOv5s algorithm and the original algorithm used the same dataset for 300 epochs of training under the same experimental environment mentioned in Section 3.1. The mean Average Precision (mAP) comparison curve of the experiments is shown in Figure 7.

As can be seen from Figure 7, after 50 epochs of training, both algorithms converged rapidly, and the improved YOLOv5s algorithm converged faster than the original algorithm. Additionally, the enhanced YOLOv5s method significantly improves the average accuracy when compared to the original algorithm.

### 3.4. Test Results

#### 3.4.1. Qualitative Analysis

In order to better show the detection results of the algorithm for the six classifications, we tested on the example diagram in Section 3.2. The detection results of the yolov5s algorithm before and after improvement are shown in Figure 8.

We can see from Figure 8 that, for the six states of helmet wearing in this research, the detection results of the original YOLOv5 algorithm missed the detection of *helmet_nowear* and falsely detected *helmet_good*, while the improved YOLOv5 algorithm could accurately detect the six states, and the confidence level was mostly higher than the original YOLOv5 algorithm. After many tests, it was found that the improved YOLOv5 algorithm had strong robustness.

In addition, in order to better test the detection effect of our algorithm on helmet wearing in a real construction scene, we selected distant and small targets, mesoscale targets, and dense targets from the test set for helmet wearing status detection. The detection results of the yolov5s algorithm before and after improvement in the real scene are shown in Figure 9.

The improved YOLOv5s algorithm had an excellent detection impact for targets at all scales and dense targets, reducing many missed and false detections, as can be observed in Figure 9. In particular, some long-distance targets at the construction site can be detected accurately, which makes the model more practical.

#### 3.4.2. Quantitative Analysis

We used the *Precision* and *Recall*, which are respectively defined in Equations (6) and (7), to quantitatively assess the performance of the model in terms of detection, and the PR curves of various categories under the YOLOv5s algorithm model before and after improvement are drawn, respectively. Figure 10 demonstrates that the enhanced YOLOv5s algorithm improved the detection performance of each classification, with the mAP improved by 3.9%.
(6)$$Precision=\frac{TP}{TP+FP}$$
(7)$$Recall=\frac{TP}{TP+FN}$$
where *TP* denotes the number of samples that predict the correct category as positive, *FP* indicates the number of samples that incorrectly predict the category as positive, and *FN* represents the number of samples that identify the correct category as negative.

From the comparison of the PR curves, it can be shown that the improved model greatly improved the detection performance of the helmet wearing state, and the improved model was particularly accurate at detecting the classes *helmet_nowear* and *hat_only*. However, due to its fine features, the class *not_fastened* is not significantly different from the classes *helmet* and *helmet_good*, and the detection performance needs to be improved. In view of the low detection accuracy of this classification, in the dataset preparation stage, we focused on supplementing and enhancing it, but the detection effect was not significantly improved. We will consider fusing the fine-grained algorithms in further research.

### 3.5. Ablation Experiment

In this research, the ablation experiments based on the YOLOv5s algorithm were designed to demonstrate the impact of each modification on the effectiveness of helmet wearing state identification more clearly. In Table 7, the experimental findings are shown.

Table 7 details the experimental results of the four improved methods mentioned in Section 2 under different combinations. Overall, the combination of various improvement methods improved the performance of helmet wearing status detection, and the four improvements together had the best effect. In the two mixed experiments for improvement, the combination of redesigning the anchor and introducing an attention mechanism had the best effect, while the combination of adding a small target layer and modifying the loss function had the worst effect. In the three mixed experiments for improvement, the combination of redesigning the anchor, adding a small target layer, and introducing an attention mechanism had the best effect, while the combination of redesigning the anchor, adding a small target layer, and modifying the loss function had the worst effect.

### 3.6. Comparative Experiment

To demonstrate the performance of the improved YOLOv5s algorithm better, we tested some highly evaluated target detection algorithms in the field of deep learning on our dataset. Table 8 shows the Average Precision (AP) of each algorithm on our six-class dataset. Table 9 compares the mAP (both IOU = 0.5 and IOU = 0.5:0.95, area = small), Frames Per Second (FPS), and the file size of each algorithm from a more macro perspective.

It can be seen from Table 8 that the improved YOLOv5s algorithm performed best in the detection of the wearing states of *helmets*, *hat_only*, *helmet_good*, and *not_fastened*; the original YOLOv5 algorithm was the top performer in the detection of *person* and *helmet_nowear*. The SSD-VGG16 algorithm performed as well as the improved YOLOv5s algorithm in detecting *not_fastened*. Table 9 shows that the improved YOLOv5s algorithm performed best in the mAP with an IOU of 0.5 and small target evaluation indicators; for the FPS, the YOLOv5s algorithm before and after the improvement had little difference, but both were much higher than the other algorithms. In addition, although the file size of the improved YOLOv5s method was 1.4 MB larger than the initial algorithm, it was still less than other competing algorithms. This feature makes the YOLOv5s algorithm have greater hardware portability and practical value.

## 4. Conclusions and Future Works

In order to solve the problem in which most of the existing helmet wearing detection algorithms only deal with whether the helmet is worn or not and do not pay attention to the various states of the helmet in the actual scene, this paper constructed a dataset with finer classification and proposed a helmet wearing state detection algorithm based on an improved YOLOv5s algorithm.

For the dataset, compared with existing datasets, the quality of the six-category dataset we built is higher, especially the added class *hat_only*, which can distinguish some cases that can be confused with class *helmet* and the class *helmet_nowear* enriches the detection capability of the model and helps in the preparation of future research. Furthermore, we made four improvements to the YOLOv5s algorithm. By adapting to the annotation of this dataset, the size of the prior box was redesigned, and a small target detection layer was added for the situation where the actual construction scene is far away and the target objects are dense. Furthermore, we introduced the attention mechanism *CoordAtt* to the algorithm and used the EIOU_Loss function to replace the original CIOU_Loss in the YOLOv5s algorithm.

According to the experiments in Section 3, the improved algorithm’s false detection and missed detection rates were lower than those of the present helmet wearing detection methods. Moreover, its detection precision and small target detection capability were greatly improved. However, our current algorithm still has some shortcomings, mainly reflected in the lack of detection accuracy of the class *not_fastened* with small differences between classes. In this study, we performed data augmentation and improved the YOLOv5s algorithm’s structure, but this did not completely solve the problem. Next, we will consider using a fine-grained algorithm to solve this problem [34,35,36]. In addition, in view of the richness and strong expression ability of our dataset, the idea for further research is to study the description of the construction images to further assist the safety monitoring of construction sites through image description.

## Figures and Tables

**Figure 1 sensors-22-09843-f001:**
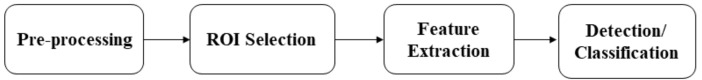
Main steps of helmet wearing detection methods based on traditional algorithms.

**Figure 2 sensors-22-09843-f002:**
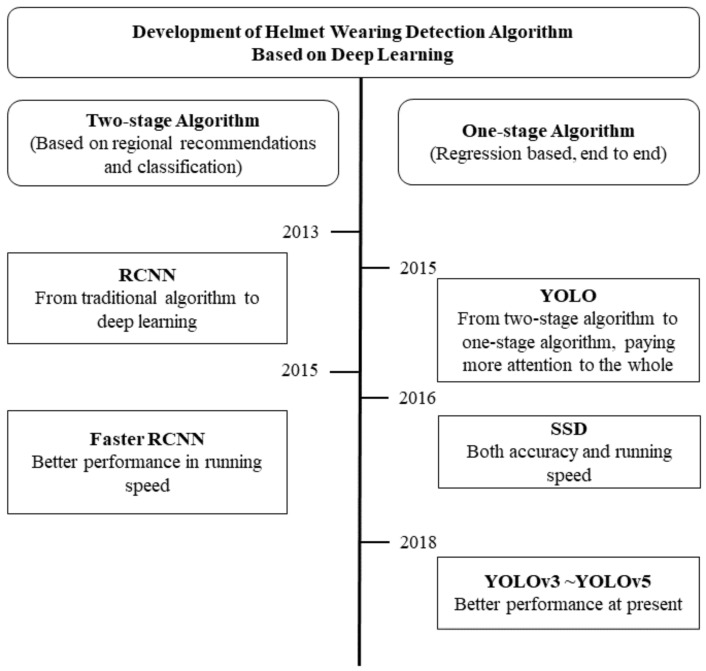
Development of helmet wearing detection algorithm based on deep learning.

**Figure 3 sensors-22-09843-f003:**
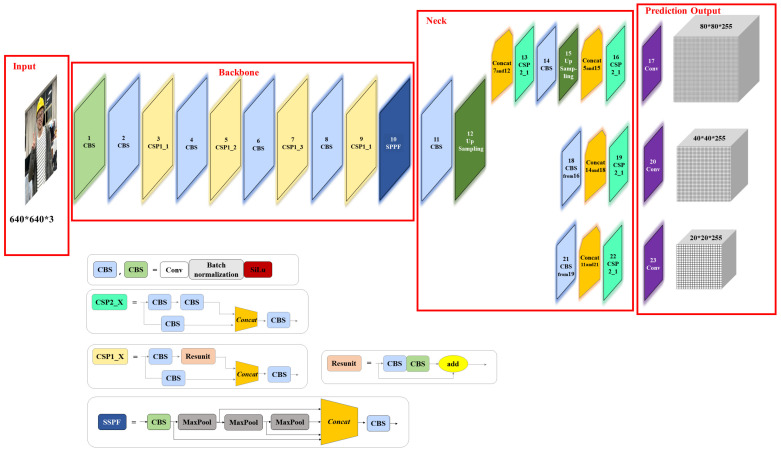
The framework of the YOLOv5s algorithm.

**Figure 4 sensors-22-09843-f004:**
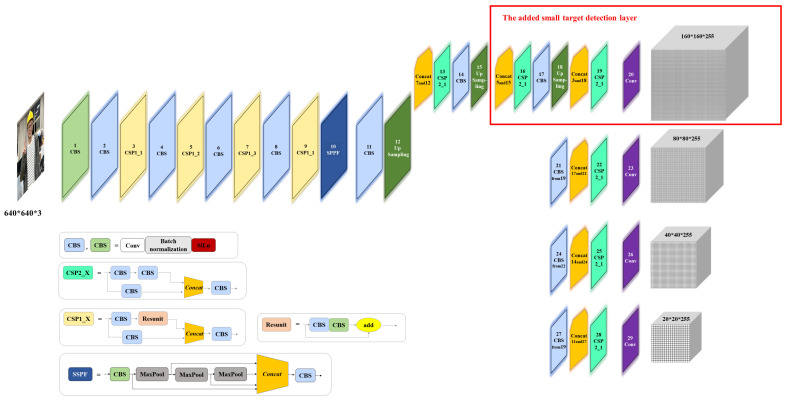
YOLOv5s algorithm framework with the added small target detection layer.

**Figure 5 sensors-22-09843-f005:**
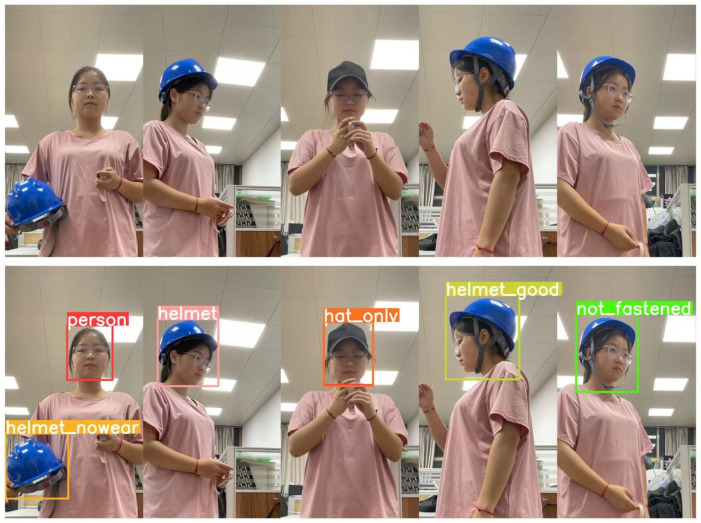
Sample diagram of the six classes.

**Figure 6 sensors-22-09843-f006:**
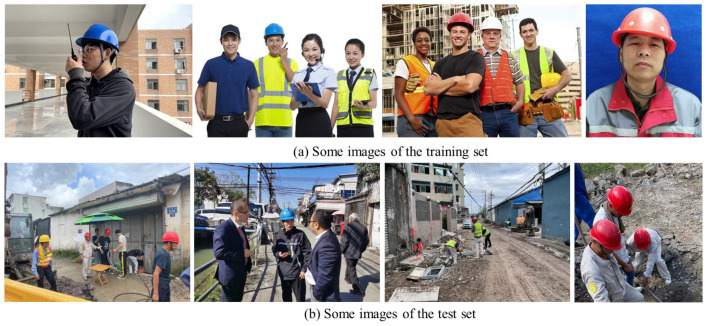
Some samples of the training set and test set.

**Figure 7 sensors-22-09843-f007:**
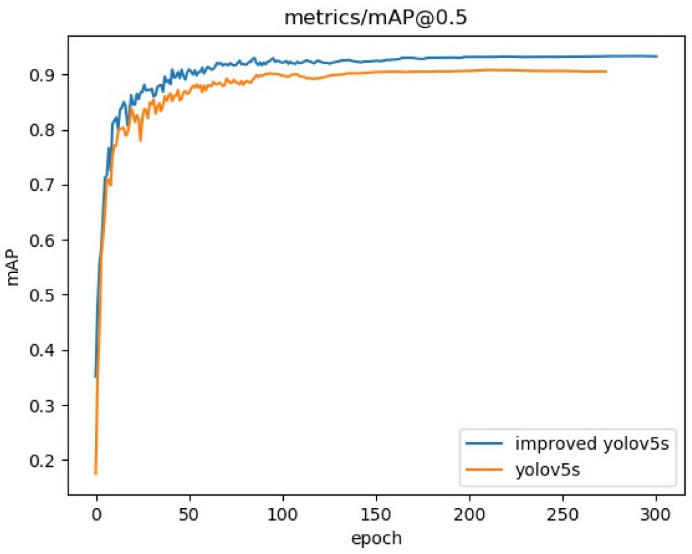
mAP@0.5 contrast curve.

**Figure 8 sensors-22-09843-f008:**
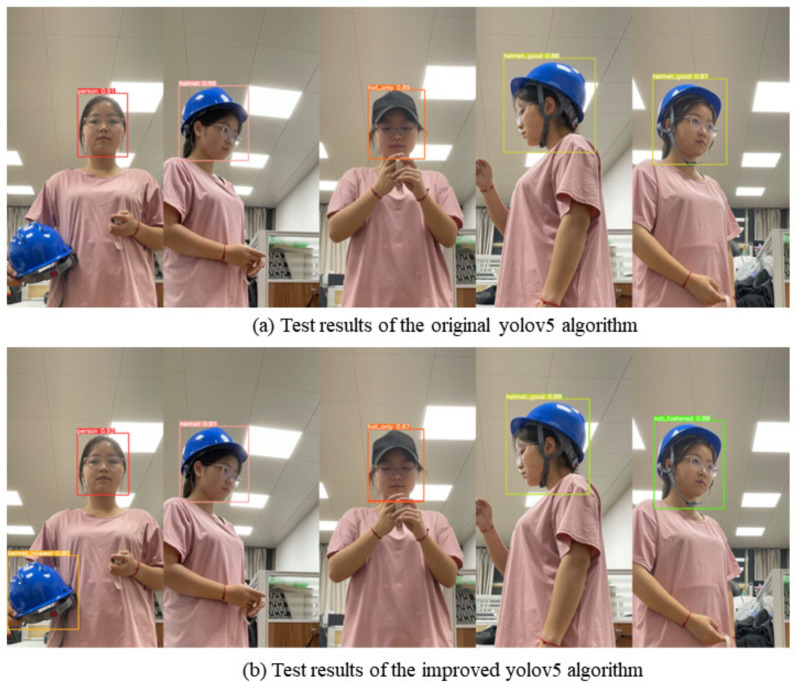
Comparison of the detection results of the six categories by the YOLOv5s algorithm before and after improvement.

**Figure 9 sensors-22-09843-f009:**
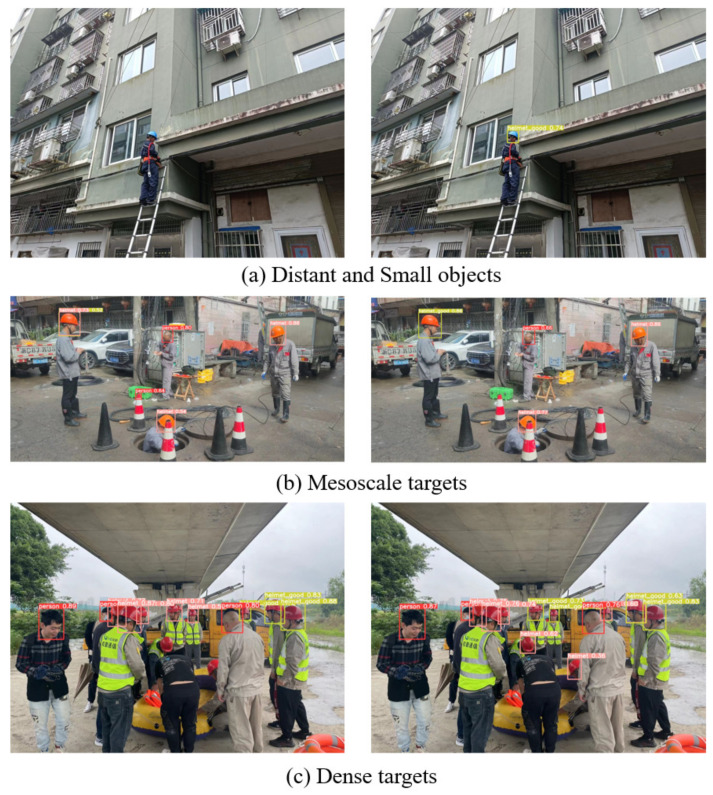
Comparison of YOLOv5s algorithm detection results before and after improvement in the real scene.

**Figure 10 sensors-22-09843-f010:**
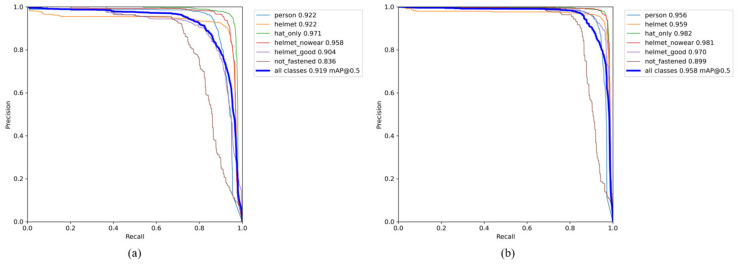
PR curve comparison. (**a**) PR curve of YOLOv5s. (**b**) PR curve of improved YOLOv5s.

**Table 1 sensors-22-09843-t001:** A priori anchor frame size before and after modification (taking the integer).

Feature Map Scale	Original Prior Box Size	Modified Prior Box Size
Add small target detection scale (160 × 160)	–	(5, 6) (8, 14) (15, 11)
Small scale (80 × 80)	(10, 13) (16, 30) (33, 23)	(15, 17) (24, 29) (39, 46)
Mesoscale (40 × 40)	(30, 61) (62, 45) (59, 119)	(63, 74) (96, 121) (156, 177)
Large scale (20 × 20)	(116, 90) (156, 198) (373, 326)	(231, 270) (385, 421) (592, 562)

**Table 2 sensors-22-09843-t002:** Comparison of convergence speed of anchor before and after modification.

Anchor	mAP@0.5 (%)	BatchSize	Epoch at End
Original Anchor	90.3	12	241
Modified Anchor	**91.4**	12	**171**
Original Anchor	90.9	32	286
Modified Anchor	**91.9**	32	**148**

**Table 3 sensors-22-09843-t003:** Partial COCO evaluation indicators.

Model	mAP@0.5 (%)	mAP@small (%)	Epoch at End (Epochs)
YOLOv5s	90.3	45.2	234
YOLOv5s with a small target layer	**91.7**	**65.6**	**199**

**Table 4 sensors-22-09843-t004:** Comparison of model mAP before and after modification of loss function.

Loss Function	mAP@0.5 (%)	Epoch at End (Epochs)
CIOU_Loss	90.3	234
EIOU_Loss	**91.5**	**191**

**Table 5 sensors-22-09843-t005:** Comparison of model mAP adding different attention modules.

Model with Attention Module	Batch_size	Epoch at End (Epochs)	mAP@0.5 (%)	Weight (Mb)
YOLOv5s	4	234	90.3	13.6
YOLOv5s + SE	4	207	92	13.7
YOLOv5s + C3SE	4	194	91.7	12.5
YOLOv5s + CBAM	4	300	91.7	13.7
YOLOv5s + ECA	4	**162**	92.1	13.6
YOLOv5s + CoordAtt	4	163	**92.3**	13.6
YOLOv5s + Transformer	4	163	92	15.9
YOLOv5s + Ghost	4	253	90.9	11.4

**Table 6 sensors-22-09843-t006:** The number of target box labels of each category in the dataset.

Dataset	Person	Helmet	hat_only	helmet_nowear	helmet_good	not_fastened
train	3299	6273	3299	1956	2218	912
val	1430	2723	1433	846	964	399

**Table 7 sensors-22-09843-t007:** Comparison of ablation experiment findings.

Model	Precision (%)	Recall (%)	mAP@0.5 (%)
YOLOv5s	90.7	84.2	90.3
^1^ Anchor **+** ^2^ Small	91.5	87.4	91.8
Anchor **+** ^3^ EIOU	90.9	86.5	91.7
Anchor **+** ^4^ Attention	**92.4**	86.5	92.4
Small **+** EIOU	90.2	85.1	90.9
Small **+** Attention	91.2	85.7	91.6
EIOU **+** Attention	92.2	86.7	92.1
Anchor **+** Small **+** EIOU	92.1	85.2	91.4
Anchor **+** Small **+** Attention	91.8	87	92.8
Anchor **+** EIOU **+** Attention	91.9	88.4	92.5
Small **+** EIOU **+** Attention	91.1	87.2	92.0
Anchor **+** Small **+** EIOU **+** Attention	**92.4**	**89.1**	**93.4**

^1^*Anchor* means to redesign the size of the prior anchor frame; ^2^
*Small* means to add a small target layer; ^3^
*EIOU* means to adopt EIOU_Loss; ^4^
*Attention* means to introduce the attention module

**Table 8 sensors-22-09843-t008:** AP of each algorithm on the six-class dataset.

Algorithms	Person	Helmet	hat_only	helmet_nowear	helmet_good	not_fastened
FasterRCNN (resnet50)	69.44%	89.73%	94.55%	93.21%	84.98%	80.17%
FasterRCNN (VGG)	65.44%	88.70%	92.41%	92.12%	79.21%	79.76%
SSD (VGG-16)	57.9%	84.37%	90.59%	90.4%	88.62%	**89.2%**
YOLOv3	83.8%	88.43%	90.92%	92.31%	64.21%	73.68%
YOLOv5s	**94.3%**	92.5%	97.8%	**94.6%**	90.4%	72.4%
Improved YOLOv5s	93.1%	**94.5%**	**98.5%**	91.7%	**93%**	**89.2%**

**Table 9 sensors-22-09843-t009:** Comparison of detection performance of each algorithm.

Algorithms	mAP@0.5 (%)	mAP@small (%)	FPS (f·s${}^{-1}$)	Weight (MB)
FasterRCNN (resnet50)	84.09	16.4	9.49	108
Faster-RCNN (VGG)	82.94	16.1	35.9	521
SSD (VGG-16)	83.51	32.3	40	93.1
YOLOv3	82.22	17	17.58	235
YOLOv5s	90.3	39.1	110	**13.6**
Improved YOLOv5s	**93.4**	**69.7**	**111**	15

## Data Availability

The study did not report any data.

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
