# Peer review of "Helmet Wearing State Detection Based on Improved Yolov5s"

_sensors, 2022, doi:10.3390/s22249843_

Round 1

Reviewer 1 Report

In this manuscript, a data set with six cases: not wearing a helmet, wearing a helmet, just wearing a hat, having a helmet but not wearing, wearing a helmet correctly and wearing a helmet without wearing the lower jaw belt, has been set up for Helmet Wearing States Detection. A modified YOLOv5s algorithm has been presented.

As shown in Figure 3, there are some logical relationships between the proposed six cases. For instance, wearing a helmet correctly and wearing a helmet without wearing the lower jaw belt both belong to the case of wearing a helmet. Should the six cases be reduced to five cases by removing that of "wearing a helmet"?

The  YOLOv5s algorithm should be properly cited, especially in Section 2.1 (there is no citation in the manuscript).

Reviewer 2 Report

Aiming at the helmet wearing detection problem, the article proposes an improved yolov5 algorithm, which studies three work contents: adding the detection layer of small target objects, introducing the attention mechanism and modifying the loss function. The article is generally logical and detailed, but the degree of innovation needs to be improved, the hat_only and the not_fastened categories need further research. In order to obtain the same results as the literature publication value, there are still some problems that need to be improved.

(1) In the second paragraph of the introduction section, various algorithms for the helmet wearing detection problem are introduced, and it is suggested to add a diagram to explain the optimization process of the helmet wearing detection algorithm.

(2) The introduction is quite simple. More vision applications in civil engineering field may be presented with state-of-art references (Novel visual crack width measurement based on backbone double-scale features for improved detection automation, Engineering Structures 2023).

(3) In Redesign the Prior Anchor Frame in Section 2.2, it is suggested to describe the work of reconstructing the prior box size using the K-means approach.

(4) In Section 2.4 Adopt EIOU_Loss, it is necessary to introduce EIOU_Loss, including but not limited to the penalty term of the loss function and the difference from CIOU_Loss.

(5) In Section 2.5 Increase Attention Mechanism, it is necessary to describe the adding position of the attention module, and it is better to give the corresponding algorithm framework diagram.

(6) In section 2.5 happens Attention Mechanism in table 5, it is suggested to indicate epochs of each model, so as to better compare the performance improvement results after the addition of different attention modules.

(7) In Section 3.2 Data Set, the format of Table 6 should be adjusted appropriately to make the display more clear and beautiful.

(8) In Section 3.3 Training Results, The mean Average Precision(MAP) comparison curve of the experiments should be annotated in Figure 5.

(9) In the Qualitative Analysis in Section 3.4.1, the test results of hat_only, helmet_nowear and not_ fastened are not compared in Figure 6, so it is recommended to add them.

(10) In Section 3.5 Ablation Experiment, there were many groups of ablation experiment, so it is recommended to display the effects of two-mixed experiment and three-mixed experiment separately in order to show them more intuitively.
